# Travel-related control measures to contain the COVID-19 pandemic: an evidence map

Ani Movsisyan [1,2] Jacob Burns [1,2] Renke Biallas,[1,2] Michaela Coenen,[1,2] Karin Geffert,[1,2] Olaf Horstick,[3] Irma Klerings,[4] Lisa Maria Pfadenhauer [1,2], Peter von Philipsborn [1,2] Kerstin Sell,[1,2] Brigitte Strahwald,[1,2] Jan M Stratil,[1,2] Stephan Voss,[1,2] Eva Rehfuess[1,2]

AM and JB are joint first authors.

[1] Institute for Medical Information Processing, Biometry and Epidemiology, Ludwig Maximilians University Munich, Munich, Germany
[2] Pettenkofer School of Public Health, Ludwig Maximilians University Munich, Munich, Germany
[3] Heidelberg Institute of Global Health, Heidelberg University, Heidelberg, Germany
[4] Department for Evidence-based Medicine and Evaluation, Danube University Krems, Krems, Austria

**Correspondence to**
Dr Ani Movsisyan;
ani.movsisyan@ibe.med.uni-muenchen.de

## ABSTRACT

**Objectives** To comprehensively map the existing evidence assessing the impact of travel-related control measures for containment of the SARS-CoV-2/COVID-19 pandemic.

**Design** Rapid evidence map.

**Data sources** MEDLINE, Embase and Web of Science, and COVID-19 specific databases offered by the US Centers for Disease Control and Prevention and the WHO.

**Eligibility criteria** We included studies in human populations susceptible to SARS-CoV-2/COVID-19, SARS-CoV-1/severe acute respiratory syndrome, Middle East respiratory syndrome coronavirus/Middle East respiratory syndrome or influenza. Interventions of interest were travel-related control measures affecting travel across national or subnational borders. Outcomes of interest included infectious disease, screening, other health, economic and social outcomes. We considered all empirical studies that quantitatively evaluate impact available in Armenian, English, French, German, Italian and Russian based on the team's language capacities.

**Data extraction and synthesis** We extracted data from included studies in a standardised manner and mapped them to a priori and (one) post hoc defined categories.

**Results** We included 122 studies assessing travel-related control measures. These studies were undertaken across the globe, most in the Western Pacific region (n=71). A large proportion of studies focused on COVID-19 (n=59), but a number of studies also examined SARS, MERS and influenza. We identified studies on border closures (n=3), entry/exit screening (n=31), travel-related quarantine (n=6), travel bans (n=8) and travel restrictions (n=25). Many addressed a bundle of travel-related control measures (n=49). Most studies assessed infectious disease (n=98) and/or screening-related (n=25) outcomes; we found only limited evidence on economic and social outcomes. Studies applied numerous methods, both inferential and descriptive in nature, ranging from simple observational methods to complex modelling techniques.

**Conclusions** We identified a heterogeneous and complex evidence base on travel-related control measures. While this map is not sufficient to assess the effectiveness of different measures, it outlines aspects regarding interventions and outcomes, as well as study methodology and reporting that could inform future research and evidence synthesis.

### Strengths and limitations of this study

► We applied systematic and standardised methods at all stages and have produced a comprehensive evidence map of travel-related control measures for containment of the SARS-CoV-2/COVID-19 pandemic.

► For title/abstract and full-text screening, we developed guidance documents and conducted piloting exercises to ensure consistency among review authors; we additionally collected and clarified all uncertainties on a rolling basis through daily team calls.

► We conducted searches in three major health-related databases and two COVID-specific databases; however, it is likely that most studies assessing economic and social outcomes are found in other databases.

► The unspecific and inconsistent reporting of primary studies with regard to interventions, especially when a package of control measures was investigated, meant that determining eligibility, as well as summarising and mapping these studies, was challenging.

► We did not include travel warning or travel advice in the evidence map, which limits the scope of this map.

## INTRODUCTION

In December 2019, the occurrence of the SARS-CoV-2 was reported in Wuhan, China. Over the next weeks, the virus and the associated respiratory disease referred to as COVID-19 spread further into China and other parts of Asia including Japan, South Korea and Thailand.[1] By mid-March 2020, when the WHO declared COVID-19 a global pandemic, cases had been observed in over 100 countries and territories across the globe.[2]

According to WHO, various travel-related control measures, such as entry/exit screening, travel bans to and/or from specific

areas within or between countries, and quarantine of travellers have since been implemented by most countries around the world to contain and mitigate the spread of SARS-CoV-2 and COVID-19.[3] Following the early-stage responses in Asian countries, strict measures, such as border closures and drastic reductions in airline travel, have been put into place in most countries around the world, starting in February 2020 and continuing into May 2020. While in the context of a rapidly evolving pandemic decisions often need to be made even in the absence of high quality evidence, efforts to identify and synthesise the best available evidence will help inform whether the currently implemented measures should be sustained, adapted or lifted. Where possible, decisions need to be based on evidence regarding the effectiveness of these measures in contributing to the control of the pandemic, as well as regarding the associated economic and social impacts. They will also need to take into account short-term and longer term costs, acceptability and feasibility of such measures.

Travel-related control measures have been assessed through systematic or narrative reviews in the context of previous epidemics and pandemics, such as influenza, severe acute respiratory syndrome (SARS-CoV-1/SARS) and Middle East respiratory syndrome (MERS-CoV/ MERS). Regarding the containment of influenza, these reviews have examined the effectiveness of a broad set of measures,[4–6] as well as the effectiveness of specific measures, such as entry/exit screening[6 7]; they have also assessed the economic implications of various pharmaceutical and non-pharmaceutical interventions during influenza pandemics.[8 9] Effectiveness of measures, such as international travel bans and entry/exit screening, have also been examined in reviews of SARS and MERS.[7 10] To date and to the best of our knowledge, the evidence on travel-related control measures for the control of the current pandemic has not been systematically assessed.

In this paper, we aim to systematically identify and map the existing evidence assessing the impact of travel-related control measures (ie, border closures, travel restrictions and bans, entry and exit screening, quarantine/isolation of travellers crossing borders and multiple interventions combined) for containment of the SARS-CoV-2/COVID-19 pandemic, drawing on evidence from the current pandemic, as well as on evidence in relation to SARS, MERS and influenza. This evidence map was commissioned by the WHO to serve as an important resource for researchers and policymakers in providing an overview of the currently available evidence in relation to various travel-related control measures to contain the COVID-19 pandemic. In itself, the evidence map is not sufficient to assess the effectiveness of different measures. Instead, it provides an important basis for decisions regarding the need for and possibility to conduct primary research as well as more specific evidence synthesis (eg, regarding a specific category of control measures or a specific type of studies).

## METHODS

### Search strategy

We designed the evidence map in accordance with the Preferred Reporting Items for Systematic Reviews and Meta-Analyses extension for Scoping Reviews (PRISMA-ScR) reporting guideline, and implemented the entire project within ten days (see online supplemental file S1) for the completed PRISMA-ScR checklist). We searched the following databases: (1) Ovid MEDLINE ALL (1946–present); (2) Embase.com (Elsevier); (3) Science Citation Index Expanded (1900–present), Social Sciences Citation Index (1900–present) and Emerging Sources Citation Index (2015–present) (Web of Science); (4) the US Centers for Disease Control and Prevention's COVID-19 Research Articles Downloadable Database (includes published articles, as well as grey literature, such as preprints) and (5) WHO COVID-19 Database (includes published articles only).

The initial search strategy was developed for MEDLINE (see online supplemental file S2) and further adapted for the other databases. All database searches were conducted up to 3 May 2020. The search strategies were designed and conducted by an experienced information specialist. The searches were conducted in English. Where database functionality allowed for it, we limited the search results to Armenian, English, French, German, Italian and Russian, based on the language capacity of the research team and considered studies for inclusion published in all of these languages. We also conducted forward and backward citation searches of all relevant (systematic) reviews identified through the database searches (see online supplemental file S3) up to 6 May 2020 in Scopus (Elsevier).

### Eligibility criteria

We determined eligibility of studies based on the investigated population/context, intervention, outcome and study type.

Regarding the population/context, this evidence map draws on direct evidence from human populations susceptible to the current SARS-CoV-2/COVID-19, as well as indirect evidence from a set of other relevant respiratory infectious diseases. To help define a set of diseases that is most similar and relevant to COVID-19, we used the following criteria: (1) diseases of viral origin; (2) mode of transmission primarily airborne via droplets/ aerosols (as well as person to person); (3) acute disease with the potential to cause an epidemic/pandemic; (4) similar clinical features (ie, non-specific febrile illness with the potential to develop into pneumonia and acute respiratory distress syndrome; difficult diagnosis based on clinical features during transmission-relevant phase, including the period prior to symptom development and during early-stage symptoms); and (5) unavailability of a vaccine and/or difficulty to contain an outbreak through vaccination. As a result, we considered:

► SARS-CoV-2/COVID-19.
► SARS-CoV-1/SARS.
► MERS-CoV/MERS.

► Influenza.

Studies in all other populations/contexts, including evidence on infectious diseases less relevant to the current SARS-CoV-2/COVID-19 pandemic (eg, avian influenza, Ebola, meningitis, HIV/AIDS, tuberculosis, dengue, plague, cholera, fever, smallpox, measles or Zika virus) were excluded.

Regarding the intervention, we considered travel-related control measures affecting human travel across national or subnational borders, specifically:

► Closure of borders (borders closed to entry and/or exit).

► Travel bans (suspension of flights, ground crossing, ship itineraries, refusal of entry or travel and visa suspension/denial) between countries and between regions and large cities within countries.

► Travel restrictions (ie, varying levels of travel reductions) between countries and between regions and large cities within countries).

► Entry/exit screening (eg, temperature measurement, health questionnaire, thermography, physical examination, laboratory tests, passive observation and/or follow-up quarantine) at airports, ports, land borders and train stations.

► Quarantine/isolation of travellers from affected regions (at borders, at designated institutions or at home).

► Any combination of the above measures and/or with other control measures (eg, combination of border closure and school closure).

All other interventions were excluded. This comprised those not directly related to travel (eg, community-based quarantine, hygiene measures and bans on mass gatherings), those related to the movement of animals or goods, travel warnings or travel advice issued by the WHO or national governments, situations affecting travel but not representing travel-related control measures (eg, school holidays) and those solely concerned with the effectiveness of laboratory tests rather than their implementation as part of an entry/exit screening procedure.

Regarding outcomes, we considered those assessing the quantitative impacts of the interventions on:

► Infectious disease outcomes (eg, number/proportion of cases, number/proportion of deaths, time to/delay in epidemic arrival or peak, reproduction number, healthcare demand and utilisation).

► Screening outcomes (eg, number/proportion of persons screened and number/proportion of those screened identified as cases).

► Other health outcomes (eg, psychosocial impact).

► Economic outcomes (eg, travel volumes, costs of measures implemented and losses to different economic sectors).

► Social outcomes (eg, stigmatisation/discrimination of foreigners, xenophobia and migration volumes).

All other outcomes, such as those on the human rights and legal implications of interventions, were excluded.

Regarding study types, we considered all types of empirical studies that quantitatively evaluate impact (eg, epidemiological, modelling, simulations and econometric studies). All other study types, including qualitative studies, diagnostic studies focused on test performance and non-empirical studies (eg, commentaries, narrative and systematic reviews) were excluded.

## Study selection

After deduplication, all titles and abstracts were screened by one reviewer (shared among several team members), excluding only those studies that were clearly irrelevant. For all studies deemed potentially relevant or unclear at the title/abstract screening stage, one reviewer screened the full text. At this stage, a final decision regarding eligibility was made. We adopted a very inclusive approach: any unclear cases were discussed with a second reviewer, and remaining uncertainties were resolved with involvement of a third reviewer and/or the whole review team. In addition, all studies excluded at the full-text screening stage based on the intervention (ie, not addressing travel-related control measures) were double-checked by a second reviewer to make sure that no relevant studies were excluded.

We used Endnote to manage collection and deduplication of records. For title and abstract screening, we used Rayyan, a web-based application designed for citation screening for systematic reviews.[11] We documented reasons for the exclusion of full texts and reported those using Microsoft Excel.

For both the title/abstract and full-text screening stages, we developed screening guidance forms to ensure that all reviewers screen similarly and consistently. We discussed inconsistencies and challenges encountered within the review team, after having screened approximately 300 titles/abstracts and 50 full texts and subsequently refined the screening guidance. We additionally collected and clarified all uncertainties in screening on a rolling basis. These were discussed in daily online meetings to ensure consistency in screening across multiple reviewers and that any questions and comments were addressed.

## Data extraction

One reviewer extracted study characteristics and data into the predefined categories of the data extraction form in Microsoft Excel (see online supplemental file S4). The extraction form was pilot-tested in the review team. The following categories were covered by the extraction form: population, setting and context; characterisation of the respiratory pathogen/disease (ie, SARS-CoV-2/COVID-19, SARS CoV-1/SARS, MERS CoV/MERS and influenza), types of interventions (eg, entry/exit screening, border closure, quarantine of travellers and travel bans), comparisons (where available), outcomes of interest (eg, health, economic and social impact) and study designs (eg, epidemiological study and modelling study).

## Mapping

We charted the extracted data based on categories and present findings in a tabular, narrative or graphical manner. Most of these categories were defined a priori, while one category, namely, geographical setting, was adapted post hoc. Specifically, we sought to define, summarise and present clusters of studies based on the pathogen/disease (ie, SARS-CoV-2/COVID-19, SARS-CoV-1/SARS, MERS-CoV/MERS and influenza), type of intervention (border closure, entry/exist screening including follow-up measures, such as isolation of positive cases, travel ban, travel-related quarantine in the absence of entry/exit screening, travel restrictions and multiple travel-related control measures combined), timing of the intervention (ie, early phase, local transmission phase, postpeak phase and unclear phase), outcomes of interest (ie, infectious disease outcomes, screening outcomes, other health outcomes, economic and social outcomes) and study designs (eg, epidemiologic study and modelling study). All data presented in the tables, text and graphics were double-checked by a second reviewer with an emphasis on accuracy in reporting on populations, interventions and outcomes and in relation to consistency of presentation.

## Patient and public involvement

Considering the time constraints and the nature of this research (ie, a rapid systematic evidence map), it was conducted without patient and public involvement.

## RESULTS

Database searches yielded a total of 4928 unique records. Through snowballing of reviews identified through the database searches, we identified an additional 1700 studies. During the title/abstract screening stage, we excluded 4445 records as clearly irrelevant. We subsequently assessed the full texts of 483 records and excluded another 361 records. Overall, we included 122 studies in this evidence map (see online supplemental file S5 for the full list). Of these, 80 were journal articles, 41 were preprints and 1 was a report. Figure 1 provides an overview of our searching and screening procedures.

## Characteristics of included studies

The 122 included studies were characterised by substantial heterogeneity in relation to the countries and populations targeted, the diseases addressed, the types of travel-related control measures examined (sometimes assessed against a range of other interventions to contain an epidemic/pandemic) and the numerous outcomes assessed. They comprise both studies that are inferential in nature and studies that are descriptive in nature, varying greatly in the specific methods applied; notably, many of the studies addressing COVID-19 were preprints. We summarise these aspects below and provide a description of each study in table 1.

## Countries and settings

Included studies assessed interventions across the globe, including, according to WHO world regions, from the Western Pacific Region (WPR) (n=71), the Southeast Asia region (n=5), the region of the Americas (n=12), the European region (EUR) (n=8), and the African region (AFR) (n=2). None of the studies included in this evidence map was conducted in a country of the Eastern Mediterranean Region (EMR). The specific countries in which interventions were implemented are listed in table 1 (under

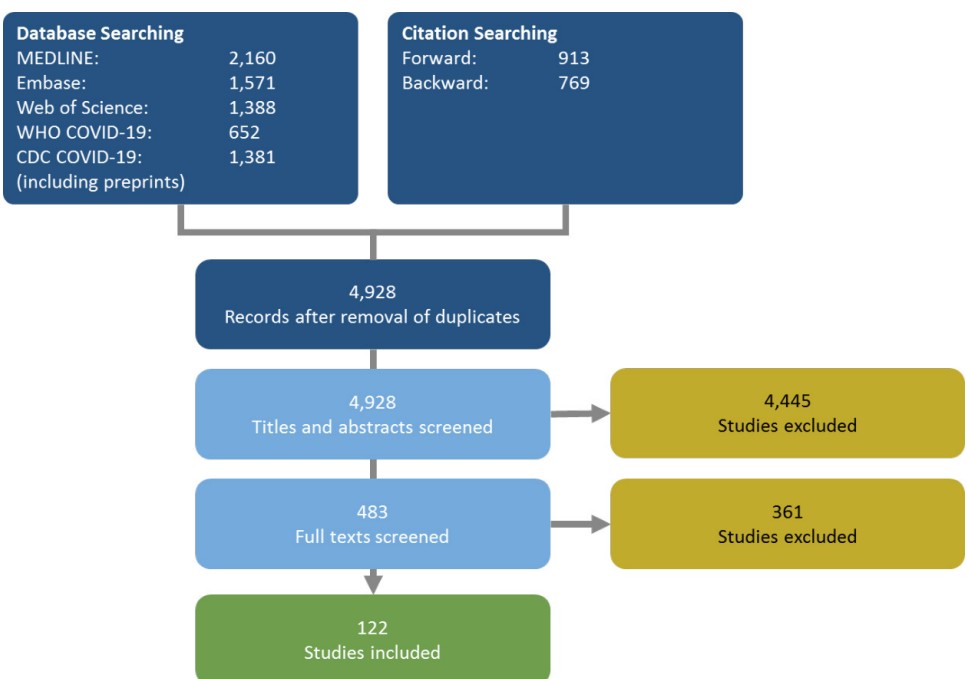

**Figure 1** Flow chart of studies identified and included during different stages of searching and screening.

**Table 1** Characteristics of included studies

| Study | Study type | WHO region | Geographical setting | Region protected by travel-related measure | Region restricted by travel-related measure | Disease | Intervention category | Outcome type |
|---|---|---|---|---|---|---|---|---|
| Adekunle 2020[14] (preprint) | Inferential | WPR | Island state | Australia | China, Iran, South Korea and Italy | COVID-19 | Travel ban | Number or proportion of cases |
| Aleta 2020[15] (preprint) | Inferential | WPR | Non-island state | Regions within China | Wuhan | COVID-19 | Multiple travel-related control measures | Number or proportion of cases |
| Anonymous 2003[16] | Descriptive | AMR | Non-island state | Vancouver, Toronto, other parts of Canada | All other countries | SARS | Entry/exit screening | Detection of high risk persons or cases |
| Anzai 2020[17] | Inferential | WPR | Non-island state | Multiple countries | China | COVID-19 | Multiple travel-related control measures | Number or proportion of cases; temporal development of epidemic |
| Arima 2020[18] | Descriptive | WPR | Island state | Japan | China | COVID-19 | Entry/exit screening; travel-related quarantine | Detection of high risk persons or cases |
| Arino 2007[19] | Inferential | n.a. | Hypothetical | Hypothetical | Hypothetical | Hypothetical infectious disease | Travel restriction | Number or proportion of cases |
| Bajardi 2011[20] | Inferential | All | Non-island state | All countries | Mexico | Influenza (H1N1 2009 pandemic) | Travel ban; travel restriction: land | Temporal development of epidemic |
| Banholzer 2020[21] (preprint) | Inferential | EUR, WPR and AMR | Non-island state | Austria, Australia, Belgium, Canada Denmark, France, Finland, Italy, Norway, Switzerland, Sweden and USA | All other countries | COVID-19 | Border closure | Number or proportion of cases |
| Bolton 2012[22] | Inferential | WPR | Non-island state | Mongolia | Neighbouring countries | Influenza (H1N1 2009 pandemic) | Travel restriction | Number or proportion of cases |
| Boyd 2017[23] | Inferential | WPR | Island state | New Zealand | All other countries | Hypothetical infectious disease | Border closure | Costs |
| Boyd 2018[24] | Inferential | WPR | Island state | New Zealand | All other countries | Hypothetical infectious disease | Border closure | Costs |
| Caley 2007[25] | Inferential | n.a. | Hypothetical | Hypothetical | Hypothetical | Influenza | Entry/exit screening; travel restriction: air | Detection of high risk persons or cases; temporal development of epidemic |

Continued

**Table 1** Continued

| Study | Study type | WHO region | Geographical setting | Region protected by travel-related measure | Region restricted by travel-related measure | Disease | Intervention category | Outcome type |
|---|---|---|---|---|---|---|---|---|
| Chang 2020[26] (preprint) | Inferential | WPR | Island state | Taiwan | Different cities in Taiwan | COVID-19 | Travel restriction | Number or proportion of cases; temporal development of epidemic |
| Cheng 2020[27] | Descriptive | WPR | Island state | Taiwan | China | COVID-19 | Exit/entry screening; travel-related quarantine | Number or proportion of cases |
| Chinazzi 2020[28] | Inferential | WPR | Non-island state | Other regions of China, all other countries | Wuhan | COVID-19 | Multiple travel-related control measures | Number or proportion of cases |
| Chiyomaru 2020[29] (preprint) | Inferential | WPR | Non-island state | Multiple countries | Multiple countries | COVID-19 | Travel ban | Number or proportion of cases |
| Chong 2012[30] | Inferential | WPR | Quasi-island state | Hong Kong | All other countries | Influenza (H1N1 2009 pandemic) | Travel restriction: land, air and maritime | Number or proportion of cases; temporal development of epidemic |
| Chung 2015[31] | Inferential | | Non-island state | Multiple countries | Multiple countries | SARS; 2006 Avian influenza; 2009 Swine influenza H1N1 | Travel restriction: air | Industry impact |
| Ciofi 2008[32] | Inferential | EUR | Non-island state | Italy | Italy and all other countries | Influenza | Travel restriction: air | Number or proportion of cases; temporal development of epidemic |
| Clifford 2020[33] (preprint) | Inferential | n.a. | Hypothetical | Hypothetical | All other countries | COVID-19 | Entry/exit screening | Detection of high risk persons or cases; temporal development of epidemic |
| Colizza 2007[34] | Inferential | All | Non-island state | All countries | All countries | Influenza | Travel restriction: air | Number or proportion of cases; temporal development of epidemic |
| Cooper 2006[35] | Inferential | All | Non-island state | All countries | All countries | Influenza | Travel restriction: air | Temporal development of epidemic |

Continued

**Table 1** Continued

| Study | Study type | WHO region | Geographical setting | Region protected by travel-related measure | Region restricted by travel-related measure | Disease | Intervention category | Outcome type |
|---|---|---|---|---|---|---|---|---|
| Costantino 2020[36] (preprint) | Inferential | WPR | Island state | Australia | All other countries | COVID-19 | Travel ban | Number or proportion of cases; number or proportion of deaths; temporal development of epidemic |
| Cowling 2020[37] | Inferential | WPR | Quasi-island state | Hong Kong | China, South Korea, Iran, Italy and affected regions in France, Germany, Japan, and Spain, Schengen Area, Macau and Taiwan. | COVID-19 | Travel restriction: air and land; travel ban; travel-related quarantine; border closure | Reproduction number; number or proportion of cases; acceptability |
| Dandekar 2020[38] (preprint) | Inferential | WPR, EUR and AMR | Island state; quasi-island state | China, Italy, South Korea and USA | All other countries | COVID-19 | Multiple travel-related control measures | Reproduction number |
| de Vlas 2009[39] | Descriptive | WPR | Non-island state | Regions of China | Regions of China | SARS | Travel-related quarantine | Reproduction number |
| Ediriweera 2020[40] (preprint) | Inferential | SEAR | Island state | Sri Lanka | All other countries | COVID-19 | Multiple travel-related control measures; travel-related quarantine | Number or proportion of cases; healthcare resources |
| Eichner 2009[41] | Inferential | WPR | Hypothetical | Hypothetical | All other countries | Influenza (pandemic) | Travel restriction: air | Probability of epidemic |
| Epstein 2007[42] | Inferential | All | Non-island state | All countries (focus on USA) | All countries | Influenza | Travel restriction: air | Number or proportion of cases; temporal development of epidemic |
| Fang 2020[43] (preprint) | Inferential | WPR | Non-island state | Other regions of China | Wuhan | COVID-19 | Multiple travel-related control measures | Number or proportion of cases |
| Ferguson 2006[44] | Inferential | AMR | Island state; quasi-island state | USA and UK | Countries connected by air traffic | Influenza (pandemic) | Border controls; travel restrictios | Number or proportion of cases; temporal development of epidemic |

Continued

**Table 1** Continued

| Study | Study type | WHO region | Geographical setting | Region protected by travel-related measure | Region restricted by travel-related measure | Disease | Intervention category | Outcome type |
|---|---|---|---|---|---|---|---|---|
| Flahault 2006[45] | Inferential | n.a. | Quasi-island state | 52 international cities and surrounding regions | Cities connected via air traffic | Influenza (pandemic) | Travel restriction: air | Number or proportion of cases |
| Fujita 2011[46] | Inferential | WPR | Island state | Regions in Japan | USA, Canada and Mexico | Influenza (H1N1 2009 pandemic) | Entry/exit screening | Detection of high risk persons or cases |
| Germann 2006[47] | Inferential | AMR | Non-island state | USA | | Influenza (pandemic) | Travel restriction: air | Number or proportion of cases |
| Glass 2006[48] | Inferential | n.a. | Non-island state | Non-infected region | Non-infected region | SARS | Entry/exit screening | Probability of epidemic |
| Gostic 2020[49] | Inferential | All | Island state | All countries | All countries | COVID-19 | Entry/exit screening | Detection of high risk persons or cases |
| Gostic 2015[50] | Inferential | n.a. | Hypothetical | Hypothetical | Hypothetical | SARS; MERS; influenza (H1N1 2009 pandemic) | Entry/exit screening | Detection of high-risk persons or cases |
| Goubar 2009[51] | Inferential | EUR | Island state; quasi-island state | Germany and UK | China and Hong Kong | SARS | Entry/exit screening | Number or proportion of cases |
| Gunaratnam 2014[52] | Descriptive | WPR | Island state | Australia | All other countries | Influenza (H1N1 2009 pandemic) | Entry/exit screening | Detection of high-risk persons or cases |
| Hale 2012[53] | Descriptive | WPR | Island state | New Zealand | Initially USA, Canada, Mexico, then all other countries | Influenza (H1N1 2009 pandemic) | Entry/exit screening | Detection of high-risk persons or cases |
| Hamidouche 2020[54] (preprint) | Inferential | AFR | Non-island state | Algeria | All neighbouring and connected (via air) countries | COVID-19 | Border closure; travel ban; travel-related quarantine | Number or proportion of cases; reproduction number |
| He 2020[55] (preprint) | Inferential | WPR | Island state; quasi-island state | China, South Korea, Italy and Iran | Italy (Lombardy, Veneto, Emilia-Romagna, Piedmont and Marche) and China (Zhejiang). | COVID-19 | Multiple travel-related control measures | Number or proportion of cases; temporal development of epidemic |
| Hien 2010[56] | Inferential | WPR | Non-island state | Vietnam | Countries with confirmed cases of 2009H1N1 | Influenza (H1N1 2009 pandemic) | Entry/exit screening | Detection of high-risk persons or cases |

Continued

**Table 1** Continued

| Study | Study type | WHO region | Geographical setting | Region protected by travel-related measure | Region restricted by travel-related measure | Disease | Intervention category | Outcome type |
|---|---|---|---|---|---|---|---|---|
| Hollingsworth 2006[57] | Inferential | n.a. | Hypothetical | Hypothetical | Countries connected via air traffic | SARS and influenza | Travel restriction: air | Number or proportion of cases; temporal development of epidemic |
| Hossain 2020[58] (preprint) | Inferential | WPR | Non-island state | Other regions of China | Wuhan | COVID-19 | Multiple travel-related control measures | Temporal development of epidemic |
| Hou 2020[59] (preprint) | Inferential | WPR | Non-island state | Other regions of China | Wuhan | COVID-19 | Multiple travel-related control measures | Number or proportion of cases; reproduction number |
| Hsieh 2007[60] | Inferential | n.a. | Hypothetical | Hypothetical | Hypothetical | Influenza (endemic) | Travel ban | Reproduction number |
| Hsieh 2006[61] | Inferential | WPR | Island state | Taiwan | Countries with SARS cases | SARS | Travel-related quarantine | Number or proportion of cases |
| Jia 2020[62] | Inferential | WPR | Non-island state | Other regions of China | Wuhan | COVID-19 | Multiple travel-related control measures | Number or proportion of cases; other |
| Jiang 2020[63] (preprint) | Inferential | WPR | Non-island state | Regions of China | Regions of China | COVID-19 | Multiple travel-related control measures | Other |
| Kerneis 2008[64] | Inferential | n.a. | Non-island state | 52 international cities and surrounding regions | 52 international cities and surrounding regions | Influenza (pandemic) | Travel restriction: air | Number or proportion of cases; temporal development of epidemic |
| Khan 2013[65] | Inferential | AMR | Non-island state | Other countries | Mexico | Influenza (H1N1 2009 pandemic) | Entry/exit screening | Detection of high-risk persons or cases; other |
| Kim 2017[66] | Inferential | n.a. | Hypothetical | Hypothetical | Hypothetical | Hypothetical infectious disease | Entry/exit screening | Reproduction number |
| Kong 2020[67] (preprint) | Inferential | WPR | Non-island state | Other regions of China | Wuhan and other cities in Hubei | COVID-19 and influenza | Multiple travel-related control measures | Number or proportion of cases |
| Kraemer 2020[68] | Inferential | WPR | Non-island state | Other regions of China | Wuhan and other cities in Hubei | COVID-19 | Multiple travel-related control measures | Number or proportion of cases |
| Kuo 2009[69] | Descriptive | WPR | Island state | Taiwan | All affected countries (eg, Mexico, USA and Canada) | Influenza (H1N2 2009 pandemic) | Entry/exit screening | Detection of high-risk persons or cases |
| Lai 2020[70] (preprint) | Inferential | WPR | Non-island state | Other regions of China | Wuhan | COVID-19 | Multiple travel-related control measures | Number or proportion of cases |

Continued

**Table 1** Continued

| Study | Study type | WHO region | Geographical setting | Region protected by travel-related measure | Region restricted by travel-related measure | Disease | Intervention category | Outcome type |
|---|---|---|---|---|---|---|---|---|
| Lam 2011[71] | Inferential | WPR | Quasi-island state | Hong Kong | All other countries | Influenza (H1N1 2009 pandemic) | Travel restriction | Probability of epidemic; temporal development of epidemic |
| Lau 2020[72] (preprint) | Inferential | WPR | Non-island state | Other regions of China | Wuhan and other cities in Hubei | COVID-19 | Multiple travel-related control measures | Number or proportion of cases; temporal development of epidemic |
| Lee 2012[73] | Inferential | n.a. | Hypothetical | Hypothetical | Hypothetical | Influenza (H1N1 2009 pandemic) | Travel restriction | Temporal development of epidemic |
| Li 2020[74] (preprint) | Inferential | WPR | Non-island state | Other regions of China | Wuhan and other cities in Hubei | COVID-19 | Multiple travel-related control measures | Number or proportion of cases |
| Lin 2020[75] (preprint) | Inferential | WPR | Non-island state | Other regions of China | Wuhan | COVID-19 | Multiple travel-related control measures | Reproduction number; other |
| Linka 2020[76] (preprint) | Inferential | EUR | Non-island state | All European countries | All European countries | COVID-19 | Multiple travel-related control measures | Number or proportion of cases |
| Liu 2006[77] | Inferential | n.a. | Hypothetical | Hypothetical | Hypothetical | Hypothetical infectious disease | Entry/exit screening | Reproduction number |
| Liu 2020[78] | Inferential | WPR | Non-island state | Other regions of China | Wuhan | COVID-19 | Multiple travel-related control measures | Number or proportion of cases |
| Liu 2020a[79] (preprint) | Inferential | WPR | Non-island state | Other regions of China | Cities outside of Hubei | COVID-19 | Multiple travel-related control measures | Number or proportion of cases |
| Malmberg 2020[80] (preprint) | Inferential | n.a. | Hypothetical | Hypothetical | Hypothetical | Hypothetical infectious disease | Travel restriction; travel-related quarantine | Probability of epidemic |
| Malone 2009[81] | Inferential | AMR | Non-island state | USA | Asia | Influenza (pandemic) | Entry/exit screening | Detection of high-risk persons or cases; number or proportion of cases; number of deaths |
| Mandal 2020[82] | Inferential | SEAR | Non-island state | India | China, Hong Kong, Singapore, Thailand, Japan, South Korea, Iran and Italy | COVID-19 | Entry/exit screening | Temporal development of epidemic; number or proportion of cases |

Continued

**Table 1** Continued

| Study | Study type | WHO region | Geographical setting | Region protected by travel-related measure | Region restricted by travel-related measure | Disease | Intervention category | Outcome type |
|---|---|---|---|---|---|---|---|---|
| Marcelino 2012[83] | Inferential | All | Non-island state | All countries | Mexico city | Influenza (pandemic) | Travel restriction: air | Number or proportion of cases |
| Mbuvha 2020[84] (preprint) | Inferential | AFR | Non-island state | South Africa | South Africa and the all other countries | COVID-19 | Travel ban | Number or proportion of cases; temporal development of epidemic |
| Mondal 2020[85] (preprint) | Inferential | SEAR | Non-island state | India | India | COVID-19 | Travel ban | Number or proportion of cases |
| Moriarty 2020[86] | Inferential | WPR and AMR | Non-island state | Japan, USA, home countries of cruise ship passengers | Home countries of cruise ships | COVID-19 | Travel-related quarantine | Detection of high risk persons or cases |
| Mummert 2013[87] | Inferential | AMR | Non-island state | USA | Mexico | Influenza (H1N1 2009 pandemic) | Entry/exit screening | Number or proportion of cases |
| Muraduzzaman 2018[88] | Descriptive | SEAR | Non-island state | Bangladesh | Saudi Arabia | MERS | Entry/exit screening | Detection of high-risk persons or cases |
| Nakata 2015[89] | Inferential | n.a. | Hypothetical | Hypothetical | Hypothetical | Hypothetical infectious disease | Travel restriction | Reproduction number |
| Nigmatulina 2009[90] | Inferential | n.a. | Hypothetical | Hypothetical | Hypothetical | Influenza | Travel restriction | Number or proportion of cases |
| Nishiura 2009[91] | Inferential | n.a. | Hypothetical | Hypothetical | Hypothetical | Influenza (pandemic) | Travel-related quarantine | Detection of high-risk persons or cases |
| Odendaal 2020[92] (preprint) | Inferential | AMR | Non-island state | USA | China and other affected countries | COVID-19 | Travel restriction | Number or proportion of cases |
| Pan 2020[93] (preprint) | Inferential | WPR | Non-island state | Other regions of China | Wuhan and other cities in Hubei | COVID-19 | Multiple travel-related control measures | Number or proportion of cases |
| Pan 2020a[94] | Inferential | WPR | Non-island state | China | China | COVID-19 | Multiple travel-related control measures | Number or proportion of cases; reproduction number |
| Pang 2003[95] | Descriptive | WPR | Non-island state | Beijing | All other countries | SARS | Entry/exit screening | Number or proportion of cases |

Continued

**Table 1** Continued

| Study | Study type | WHO region | Geographical setting | Region protected by travel-related measure | Region restricted by travel-related measure | Disease | Intervention category | Outcome type |
|---|---|---|---|---|---|---|---|---|
| Pinkas 2003[96] | Descriptive | EUR | Non-island state | Poland | All other countries | COVID-19 | Entry/exit screening; border closure; travel ban; travel-related quarantine | Number or proportion of cases |
| Pitman 2005[97] | Descriptive | EUR | Quasi-island state | UK | Any of the top 100 sources of international airline passengers | SARS and influenza | Entry/exit screening | Detection of high-risk persons or cases |
| Priest 2013[98] | Descriptive | WPR | Island state | New Zealand | Australia | Influenza (type A and B) | Entry/exit screening | Detection of high-risk persons or cases |
| Pullano 2020[99] | Inferential | WPR | Non-island state | Other regions of China and European countries | Wuhan | COVID-19 | Travel ban | Probability of epidemic |
| Qiu 2020[100] (preprint) | Inferential | WPR | Non-island state | Other regions of China | Wuhan and other cities in Hubei | COVID-19 | Multiple travel-related control measures | Number or proportion of cases |
| Quilty 2020[101] (preprint) | Inferential | WPR | Non-island state | Beijing, Chongqing, Hangzhou and Shenzhen | Wuhan | COVID-19 | Multiple travel-related control measures | Number or proportion of cases; probability of epidemic |
| Ray 2020[102] | Inferential | SEAR | Non-island state | India | All other countries | COVID-19 | Travel ban | Number or proportion of cases |
| Sakaguchi 2012[103] | Descriptive | WPR | Island state | Japan | Mexico, USA and Canada | Influenza (A H1N1 2009 pandemic) | Entry/exit screening | Detection of high-risk persons or cases |
| Samaan 2004[104] | Descriptive | WPR | Island state | Australia | All countries affected by SARS | SARS | Entry/exit screening | Detection of high-risk persons or cases |
| Sang 2012[105] | Inferential | n.a. | Hypothetical | Hypothetical | Hypothetical | SARS | Entry/exit screening | Reproduction number; temporal development of epidemic; number or proportion of cases; number or proportion of deaths |
| Scala 2020[106] (preprint) | Inferential | EUR | Non-island state | Individual regions of Italy | Individual regions of Italy | COVID-19 | Travel restriction | Temporal development of epidemic |
| Scalia Tomba 2008[107] | Inferential | n.a. | Hypothetical | Hypothetical | Hypothetical | Influenza (pandemic) | Travel restriction/ border control | Temporal development of epidemic |

Continued

**Table 1** Continued

| Study | Study type | WHO region | Geographical setting | Region protected by travel-related measure | Region restricted by travel-related measure | Disease | Intervention category | Outcome type |
|---|---|---|---|---|---|---|---|---|
| Shi 2020[108] (preprint) | Inferential | WPR | Non-island state | Other regions of China | Wuhan | COVID-19 | Multiple travel-related control measures | Number or proportion of cases |
| Song 2020[109]Wang 2020[110] | Inferential | WPR | Non-island state | Other regions of China | Wuhan | COVID-19 | Multiple travel-related control measures | Number or proportion of cases; reproduction number |
| St. John 2005[111] | Descriptive | AMR | Non-island state | Canada | All other countries | SARS | Entry/exit screening | Detection of high-risk persons or cases |
| Su 2020[112] (preprint) | Inferential | WPR | Non-island state | Beijing, Shanghai, Guangzhou and Shenzhen | Beijing, Shanghai, Guangzhou and Shenzhen | COVID-19 | Multiple travel-related control measures | Number or proportion of cases; temporal development of epidemic |
| Tang 2020[113]T | Inferential | WPR | Non-island state | Beijing | Wuhan and other cities in China | COVID-19 | Multiple travel-related control measures | Number or proportion of cases |
| Tian 2020[114] (preprint) | Inferential | WPR | Non-island state | Other regions of China | Wuhan and other cities in China | COVID-19 | Multiple travel-related control measures | Number or proportion of cases; temporal development of epidemic |
| Tsuboi 2020[115] | Descriptive | WPR | Island state | Japan | Diamond Princess | COVID-19 | Travel-related quarantine | Number or proportion of quarantined diagnosed |
| Wang 2015[116] | Inferential | n.a. | Hypothetical | Hypothetical (low-risk patches) | Hypothetical (high-risk patch) | Influenza (H1N1 2009 pandemic) | Entry/exit screening | Reproduction number |
| Wang 2020[110] | Inferential | WPR | Non-island state | China | Wuhan | COVID-19 | Multiple travel-related control measures | Number or proportion of cases; number or proportion of deaths |
| Wang 2007[117] | Descriptive | WPR | Island state | Taiwan | All SARS-affected regions | SARS | Travel-related quarantine | Detection of high-risk persons or cases diagnosed |
| Wang 2012[118] | Inferential | WPR | Non-island state | Different regions in China | Different regions in China | Influenza (pandemic) | Travel restriction | Temporal development of epidemic |
| Wells 2020[119] | Inferential | WPR | Non-island state | All other countries | China | COVID-19 | Multiple travel-related control measures | Number or proportion of cases; probability of epidemic |
| Weng 2015[120] | Inferential | WPR | Hypothetical | Hypothetical | Hypothetical | Influenza (H1N1 2009 pandemic) | Travel restriction | Number or proportion of cases |

Continued

**Table 1** Continued

| Study | Study type | WHO region | Geographical setting | Region protected by travel-related measure | Region restricted by travel-related measure | Disease | Intervention category | Outcome type |
|---|---|---|---|---|---|---|---|---|
| Wilder-Smith 2003[121] | Descriptive | WPR | Non-island state | Singapore | All SARS-affected regions | SARS | Entry/exit screening | Detection of high-risk persons or cases |
| Wood 2007[122] | Inferential | WPR | Non-island state | Melbourne and Sydney | Sydney and Darwin | Influenza | Travel restriction: air | Temporal development of epidemic |
| Yang 2020[123] | Inferential | WPR | Non-island state | Other regions of China, particularly Guangdong and Zhjiang provinces | Hubei | COVID-19 | Multiple travel-related control measures | Number or proportion of cases; temporal development of epidemic |
| Ying 2020[124] (preprint) | Inferential | WPR | Non-island state | Other regions of China | Hubei | COVID-19 | Multiple travel-related control measures | Number or proportion of cases |
| Yu 2012[125] | Inferential | WPR | Non-island state | China | All other countries | Influenza (H1N1 2009 pandemic) | Entry/exit screening | Number or proportion of cases |
| Yuan 2020[126] (preprint) | Inferential | WPR | Non-island state | Other regions of China and countries | Wuhan | COVID-19 | Multiple travel-related control measures | Temporal development of epidemic |
| Yuan 2020a[127] (preprint) | Inferential | WPR | Non-island state | China | Wuhan | COVID-19 | Multiple travel-related control measures | Number or proportion of cases |
| Zhang 2012[128] | Descriptive | WPR | Non-island state | Beijing | All other countries | Influenza (H1N1 2009 pandemic) | Entry/exit screening | Number or proportion of screened identified as cases |
| Zhang 2014[129] | Inferential | n.a. | Hypothetical | Hypothetical | Hypothetical | Influenza (pandemic) | Entry/exit screening | Number or proportion of cases; temporal development of epidemic |
| Zhang 2019[130] | Inferential | n.a. | Hypothetical | Hypothetical | Hypothetical | Influenza (H1N1 2009 pandemic) | Entry/exit screening; reductions in human mobility | Temporal development of epidemic |
| Zhang 2020[131] | Inferential | WPR | Non-island state | Other regions of China | Wuhan and other cities in Hubei | COVID-19 | Multiple travel-related control measures | Number or proportion of cases |
| Zhang 2020a[132] (preprint) | Inferential | WPR | Non-island state | Other regions of China | Wuhan and other cities in Hubei | COVID-19 | Multiple travel-related control measures | Number or proportion of cases; reproduction number |

Continued

| Study | Study type | WHO region | Geographical setting | Region protected by travel-related measure | Region restricted by travel-related measure | Disease | Intervention category | Outcome type |
|---|---|---|---|---|---|---|---|---|
| Zhao 2020[133] | Inferential | WPR | Non-island state | Regions of China | Regions of China | COVID-19 | Multiple travel-related control measures | Number or proportion of cases; reproduction number; temporal development of epidemic |
| Zhou 2020[134] (preprint) | Inferential | WPR | Non-island state | Other regions of China | Wuhan and other cities in Hubei | COVID-19 | Multiple travel-related control measures | Number or proportion of cases; number or proportion of deaths |
| Zlojutro 2019[135] | Inferential | AMR | Non-island state | USA | All other countries | Influenza (H1N1 2009 pandemic) | Entry/exit screening | Number or proportion of cases; cost; other |

**Table 1** Continued

AFR, African region; AMR, region of the Americas; EUR, European region; MERS, Middle East respiratory syndrome; SARS, severe acute respiratory syndrome; SEAR, Southeast Asia region; WPR, Western Pacific Region.

'regions protected by travel related measure'). Several modelling studies looked at hypothetical regions (n=16).

Of the identified interventions, some were implemented in island states (n=24) or in what we define quasi-island states (n=9), such as the UK or Hong Kong, which have limited land borders connected through, for example, a tunnel. Most interventions were implemented in non-island states (n=74). The remaining interventions assessed hypothetical regions, some of which were framed as comparable to island states, while others were more general in their framing and assumptions.

### COVID-19 and other relevant respiratory infectious diseases

Included studies assessed the impact of interventions aiming to prevent or slow the transmission of COVID-19 and other infectious respiratory diseases. We identified a large number of studies focusing on COVID-19 (n=59); we also found studies focusing on SARS (n=11), MERS (n=1) and various strains of influenza (n=39). Other studies looked at multiple infectious respiratory diseases (n=5) or a hypothetical infectious disease with COVID-19 relevant properties (n=7).

Figure 2 illustrates how the number of publications concerned with each disease has developed over time and explores where interventions have been implemented. As expected, there has been a rapid burst of research related to COVID-19 travel restrictions in 2020 (panel A), currently consisting mostly of non-peer-reviewed preprints. Research on SARS, MERS and influenza travel restrictions is more spread over time and also clusters around specific outbreaks (eg, SARS 2003; H1N1 influenza 2009) (panel B).

### Intervention categories and interventions

The 122 included studies assessed the impact of a wide range of travel-related control measures. We classified these according to broad intervention categories, including border closures (n=3), entry/exit screening (n=31), travel-related quarantine (n=6), travel bans, such as suspension of international flights (n=8), and travel restrictions (n=25).

For travel restrictions, a few studies described imposing restrictions in relation to the mode of travel (eg, restricting air, land or maritime travel); most used the term 'travel restrictions' without providing any specification. However, most of the studies in this category were modelling studies, which commonly simulated 'travel restrictions' as different percentage reductions in the travel volume (eg, 50% and 90%).

We identified a relatively large number of studies assessing the impact of a bundle of different travel-related control measures (eg, entry/exit screening and quarantining all arriving passengers) (n=49). More than half of these (n=29) assessed the impact of the lockdown of Wuhan (n=29), combining several travel-related measures. We classified all of these studies as assessing *multiple travel-related control measures* (see online supplemental file S6), as

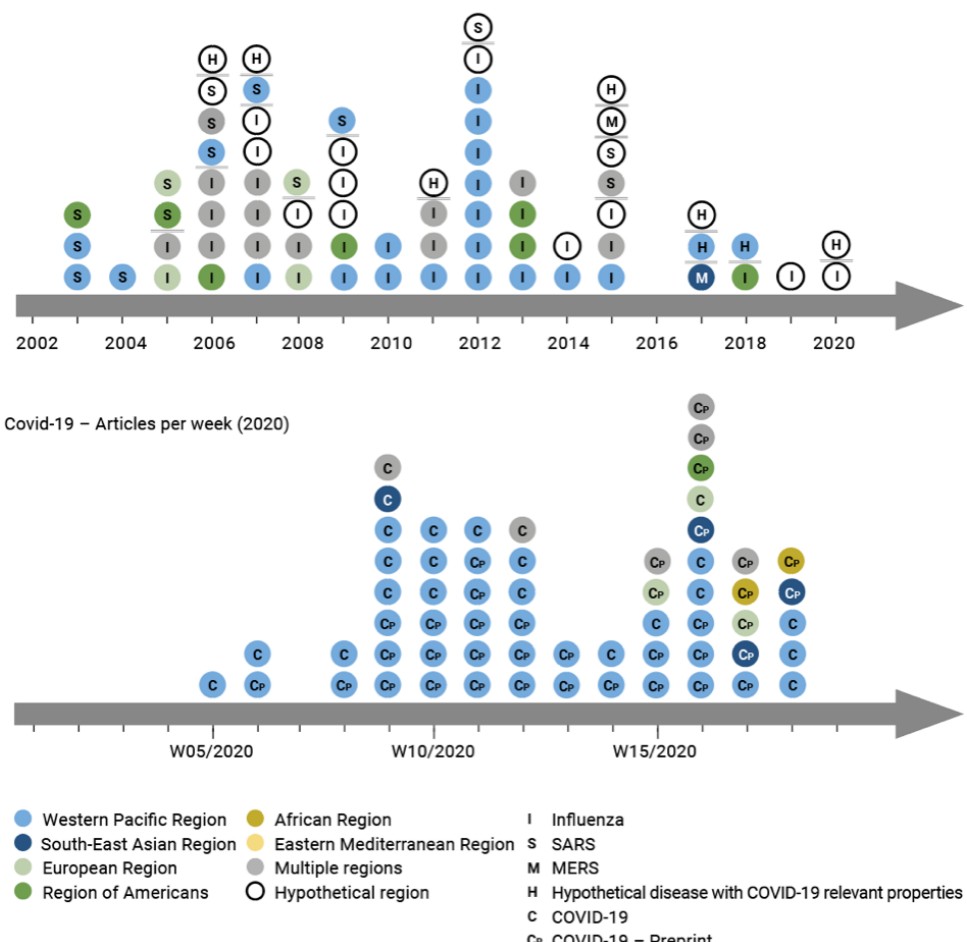

SARS, MERS, Influenza and hypothetical disease with COVID-19 relevant properties – Articles per year (2003– 2020)

Covid-19 – Articles per week (2020)

| ● Western Pacific Region | ● African Region | I Influenza |
| ● South-East Asian Region | ● Eastern Mediterranean Region | S SARS |
| ● European Region | ● Multiple regions | M MERS |
| ● Region of Americans | ○ Hypothetical region | H Hypothetical disease with COVID-19 relevant properties |
| | | C COVID-19 |
| | | Cp COVID-19 – Preprint |

**Figure 2** Illustration of the number of studies published over time; the top panel (2002–2020) shows studies focused on SARS, MERS, influenza and hypothetical disease with COVID-19 relevant properties, while the bottom panel (2020) shows studies focused on COVID-19. The specific disease is indicated by the single letter within the circle. Additionally, the colour represents the WHO world region. MERS, Middle East respiratory syndrome; SARS, severe acute respiratory syndrome.

it was impossible to identify the specific measures assessed due to lack of reporting.

Figure 3 visualises the body of evidence according to the respective disease, intervention implemented, as well as region in which an intervention was implemented. It becomes clear that the majority of the evidence assesses multiple travel-related control measures to delay or limit the progression of COVID-19. The second largest block of evidence emerges from studies focusing on various influenza strains and the impact of travel restrictions and entry/exit screening.

A critical aspect in relation to the likely impact of travel-related control measures is the timing of implementation. Figure 4 (panel A) visualises during which phase of an epidemic or pandemic different types of interventions addressed by included studies were implemented. With respect to the timing of implementation we distinguish the following:

► Early phase: interventions are implemented at a time when there are either no or only singular detected/notified cases (ie, all cases are detected

and quarantined). During this phase, imported cases represent the main source of infections.

► Local transmission phase: interventions are implemented during more or less widespread human-to-human transmission of the disease. During this phase, local transmission and imported cases represent sources of infections.

► Postpeak phase: interventions are implemented during/after successful containment of an initial outbreak/epidemic/pandemic with the possibility of recurrence. During this phase, imported cases may again represent a major source of infections.

► Unclear phase: the phase of the outbreak/epidemic/pandemic is not reported or is not directly relevant for implementation of the intervention.

Figure 4 (panel A) shows, for example, that many interventions, especially those comprising multiple measures, are implemented when local transmission of the disease has already been established. Travel restrictions are often employed in the early phase of an epidemic or pandemic while entry/exit screening measures tend

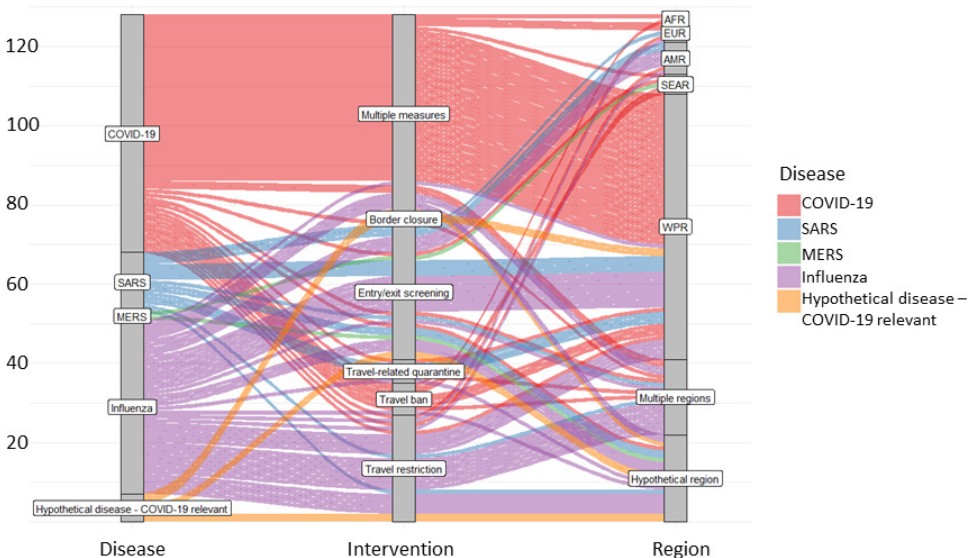

**Figure 3** Overview of the body of evidence showing the frequency of studies investigating the specific diseases (left column), interventions (middle column) and the WHO world regions (right column). MERS, Middle East respiratory syndrome; SARS, severe acute respiratory syndrome.

to be implemented both in the early stages, as well as throughout the phases of an epidemic or pandemic. Further details on the specific travel-related control measures reported in each included study and their phase of implementation is presented in the online supplemental file S6.

### Outcome categories and outcomes

We considered studies assessing five broad categories of outcomes: infectious disease outcomes, screening outcomes, other health outcomes, economic outcomes and social outcomes. Identified studies, however, largely assessed infectious disease (n=98) and/or screening outcomes (n=25). We identified no studies concerned with other health outcomes and very few studies assessing economic (n=5) or social outcomes (n=1).

Infectious disease outcomes included several types of outcomes related to disease timing and transmission, including the number or proportion of cases, the number or proportion of deaths, the reproduction number, the probability of an epidemic, demand for healthcare resources and the temporal development of the epidemic. Studies assessed various specific outcomes under these broader types, for example, 'time to epidemic peak' and 'delay of epidemic' both belong to the outcome type 'temporal development of the epidemic'. Screening outcomes all comprised some form of measuring the number and/or proportion of high-risk persons and/or cases detected. Economic outcomes covered costs and industry impact, and social outcomes examined the acceptability of travel-related control measures. These outcome types as assessed in each study are listed in table 1. A comprehensive list of the specific outcomes reported for each category and type of outcome and how often these were used across the evidence map can be found in online supplemental file S7.

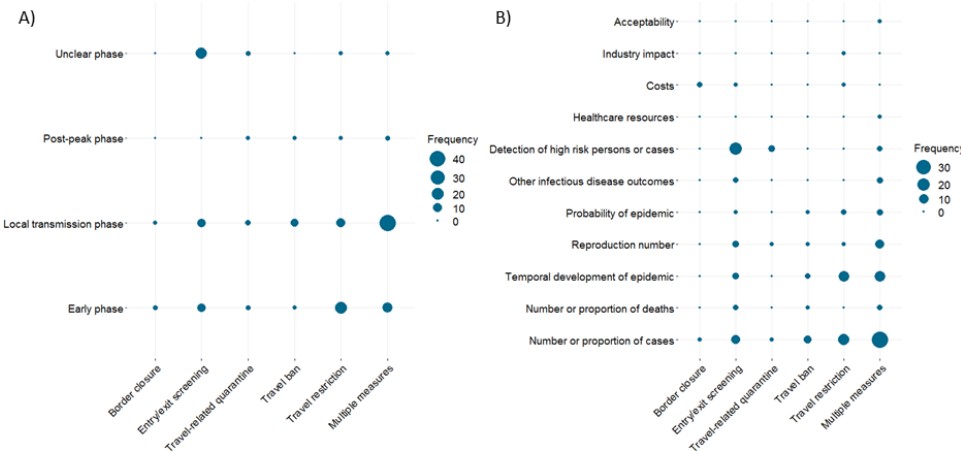

**Figure 4** Bubble plots illustrating in included studies (A) during which phase of an epidemic or pandemic different types of interventions were implemented and (B) which intervention categories were assessed against different types of outcomes.

Figure 4 (panel B) illustrates which intervention categories were assessed by different outcome categories. Infectious disease outcomes such as the number or proportion of cases, the temporal development of an epidemic and the reproduction number were well represented across all intervention types. Conversely, screening outcomes were assessed primarily with respect to entry/exit screening. The lack of studies assessing economic and social outcomes is also clearly visible.

## Study types

We included all studies assessing the quantitative impact of travel-related control measures on infectious disease, screening, economic and social outcomes. As a consequence of this broad scope, included studies employed a range of vastly different approaches drawing from the fields of infectious disease research, epidemiology, economics, biology and mathematics, among others. We categorised each study as being either inferential or descriptive. Most of the studies we found were inferential in nature (n=103), aiming to retrospectively calculate or prospectively forecast the impact of one or multiple travel-related control measures on outcomes. The remaining studies were descriptive in nature (n=19), aiming to describe the impact of control measures through summary statistics and/or graphics.

Within these broad categories, however, included studies varied greatly with regard to the specific approach taken. Inferential studies applied numerous modelling and epidemiological techniques; compartmental models, such as SIR models (S: susceptible, I: infectious, R: recovered) or SIR model derivatives, were common. Several studies also applied spatial models to explore how disease transmission moves geographically. Epidemiological time series models, as well as other epidemiological modelling and testing strategies, were also common. Descriptive studies comprised primarily observational studies and graphical summary studies, both of which measured and reported descriptive summary statistics related to intervention impact. The types of studies illustrated here are not exhaustive, and the methodological boundaries between the approaches employed are sometimes blurry. In fact, several studies apply multiple techniques, combining, for example, compartmental and spatial models or compartmental and time series models. online supplemental file S8 provides a more detailed overview of the study types included, along with examples from the included studies.

## DISCUSSION
## Summary of findings

This evidence map provides a comprehensive overview of travel-related control measures available for the control of SARS-CoV-2/COVID-19.

Most of the included studies used infectious disease or epidemiological modelling methods to examine the impact of travel-related control measures on the current COVID-19 pandemic. We also identified studies on SARS-CoV-1/SARS and influenza, mostly undertaken in the context of previous epidemics/pandemics. We found very few studies addressing MERS-CoV/MERS. The identified studies assessing travel-related control measures in the context of SARS-CoV-2/COVID-19, were mostly preprint publications that have not yet undergone peer review and are often characterised by poor reporting and conduct.[12]

Studies were undertaken across the globe. The geographical region most represented was WPR, driven in part by a number of studies focusing on the Hubei region of China and in part by studies across the region during the previous SARS-CoV-1/SARS outbreak. AFR and EUR were the least represented; we did not identify any studies from EMR. Most studies operated at the level of countries and reported little about specific settings of interest (eg, a named airport and a specific country border) or the broader implementation context (eg, usual border arrangements when control measures are not in place).

Travel-related control measures can be classified as (1) border closure, (2) travel bans, (3) travel restrictions, (4) entry/exit screening, (5) travel-related quarantine and (6) multiple travel-related control measures. Studies were identified and mapped for all of these categories. We identified a relatively large body of evidence on entry/exit screening, as well as on travel restrictions. The latter also included subnational measures, for example, city-to-city travel restrictions across broader regions, making a clear distinction from general social distancing measures challenging. We also found many studies examining the impact of bundles of travel-related control measures. Interventions are often poorly described, both in relation to the measure itself (eg, border closure) and in relation to how the measure is implemented or enforced (eg, border patrols, fines and exceptions). Moreover, travel-related control measures rarely happen in a vacuum: the closer to real-life the intervention (and the study), the more cointerventions tend to be involved. This makes it very difficult to assess the unique impact of a specific measure.

The impact of travel-related control measures in controlling geographical spread and overall disease transmission likely varies between early, local transmission and postpeak phases of an outbreak/epidemic/pandemic, yet reporting of the timing of implementation tends to be poor. Most studies focused on the early and local transmission phases; few studies were concerned with the postpeak phase.

The identified studies almost exclusively examined infectious disease and screening outcomes. Surprisingly, we did not identify any study reporting on other health outcomes, such as implications for the physical and psychosocial health of stranded travellers, of those unable to visit family members, of regular commuters unable to reach their workplace, of individuals quarantined, of people unable to obtain medical treatment and other collateral damage (eg, suspended immunisation programmes and impacts on food supply due to

limited air, maritime and land travel of either people or goods). Only three included studies were concerned with economic and social outcomes.

The bulk of the evidence derives from modelling studies. In an acute outbreak situation, the time and resources that can be dedicated to conducting timely, empirical research are limited. Modelling studies vary greatly in type (eg, SIR-like models vs time series models) and in the underlying assumptions regarding the disease, interventions or regions/settings. Additionally, many studies do not fall into clean categories, and a single study may combine compartmental infectious disease models with epidemiological spatial models. They also vary with respect to whether and how they have been validated through real-life applications.

## Gaps in the current evidence base

The evidence map identified several gaps in the evidence base related to travel-related control measures that could inform future research as well as evidence synthesis. First, some regions were under-represented, specifically studies conducted or simulated for the countries in the AFR, EMR and EUR were lacking. Second, the most commonly reported travel-related control measures were entry/exit screening and restrictions, which in the context of modelling studies were simulated as different levels of travel reductions. Thus, the evidence base regarding more stringent measures, such as travel bans and complete border closures, remains sparse. Third, our evidence map identified lack of consideration of the impact of these measures on broader health outcomes, such as physical and mental health, as well as social and economic outcomes, such as cost and burden on communities and socioeconomic inequalities. Finally, the key gap relates to the lack of empirical studies assessing the impact of travel-related control measures, including experimental and quasiexperimental approaches.

## Methodological limitations of this study

This evidence map was put together over 10 days, and the process is thus characterised by several limitations.

While we conducted searches in three major databases and two COVID-specific databases, these were mostly health centric. We only searched Web of Science—comprising Science Citation Index Expanded (1900–present), Social Sciences Citation Index (1900–present) and Emerging Sources Citation Index (2015–present)—to identify social and economic studies. As we identified very few economic and social outcomes, it is likely that there is another body of evidence to be located with more focused searches in specific economic and social science databases. In addition, we did not search for grey literature sources. Furthermore, since COVID-19 initially started in Wuhan, China, many studies are published on the topic in Chinese journals and databases. Because of the language barrier, we did not consider these sources.

The unspecific and inconsistent reporting of primary studies with regard to interventions, especially when a package of control measures was investigated, sometimes made inclusion/exclusion decisions difficult. While we developed and calibrated screening guidance to ensure consistency among reviewers, it is nevertheless possible that we excluded some travel-related control measures that were not explicitly described as such. Also, we did not undertake double-screening of all studies but only reassessed a subset of studies excluded at the full-text screening stage. In general, however, we applied a very conservative approach to title/abstract, as well as full-text screening, where any uncertainties associated with a study were marked for further checking by a second reviewer and/or for discussion among the whole review team.

We developed and calibrated data extraction guidance to be used consistently by reviewers. Nevertheless, we had to refine some categories post hoc. We also had a large number of reviewers extracting data, which created heterogeneity in the dataset. To address this, a second reviewer double-checked all extracted information for each of the main domains, that is, all information regarding interventions, outcomes and study design. Overall, high-quality data extraction was limited by poor reporting of the travel-related control measures and the specific contexts in which they are implemented in primary studies.[13]

Finally, we did not include travel warning or travel advice in the evidence map, which limits the scope of this map. While these may be classified as travel-related control measures,[13] their inclusion would have likely broadened our search strategy and prolonged the timeline of its development. In general, lack of intervention specification in the included studies makes it challenging to draw clear categories of interventions without potential overlap (eg, drawing clear distinction between a travel ban vs a travel restriction and when these are used only as descriptors in the studies without further specification).

## Implications for moving forward

An evidence map is not designed to assess the effectiveness of interventions, in this case the effectiveness of travel-related control measures in controlling infectious disease spread. The present evidence map sheds light on the variety of evidence available with regards to the quantifiable impacts of travel-related control measures, the outcomes used, as well as the study types employed. It thus represents a stepping stone towards a systematic review on the effectiveness of all or a subset of travel-related control measures.

Given our health-centric searches, we feel confident that we have identified most of the available body of evidence regarding the quantifiable impacts of travel-related control measures on health. In terms of conducting a systematic review of effectiveness, it would, however, be advisable to undertake additional forward-backward citation searches and/or similar studies searches with included studies. We identified a number of challenges related to the full analysis of this evidence base. These include: (1) classifying interventions in an appropriate and consistent manner;

(2) capturing the details of interventions (eg, components, timing and implementation characteristics) and cointerventions through an analysis of linked sources of evidence; (3) assessing the quality and usefulness of different types of studies ranging from simple observational studies to complex modelling studies; (4) quantitatively synthesising this extremely heterogeneous evidence base; and (5) dealing with the large number of preprint studies and their varying quality. Moving forward with a full analysis, we suggest reviewers plan ahead and develop strategies on how these challenges can be adequately managed. For example, this might entail consideration of external sources of evidence beyond scientific databases, such as governmental websites on the implementation of different travel-related control measures (eg, their timing and duration) and other measures in local contexts. To contain the COVID-19 pandemic, governments have employed a range of public health measures often in a bundle, which challenges assessment of the effectiveness of any single measure, such as a travel ban. It would therefore be important to explicitly document and, where possible, assess various combinations of measures implemented—including those related to travel–in future research.

Importantly, decisions to maintain or stop travel-related control measures are determined by a range of factors beyond effectiveness, including legal and human rights aspects, as well as considerations of broader health, economic and social implications, as well as sociocultural and political acceptability. Gathering evidence about these broader factors in a systematic manner was beyond the scope of the current evidence map; however, such aspects might be important to inform decision making. Future evidence synthesis would therefore require a different scope and a broader search strategy, notably encompassing searches in economics and social science databases, as well as multidisciplinary databases (eg, EconLit, PsycINFO and Scopus).

**Acknowledgements** We are grateful to Susan L Norris and Carmen Dolea for their guidance in scoping this evidence map.

**Contributors** JB, AM, ER and OH defined the study scope and developed the study protocol with significant intellectual input from all review authors. JB and AM coordinated the entire study process. IK designed the search strategy and conducted all the searches. JB, AM, RB, MC, KG, LMP, PvP, KS, BS, JMS, SV and ER conducted data screening and extraction. JB, AM, JMS, LMP and PvP did data mapping. JB, AM and ER prepared the first draft of the manuscript, which was further critically examined by all the authors and revised based on their feedback. All review authors reviewed and approved the final draft.

**Funding** The conduct of this evidence map was funded by the World Health Organization (WHO; grand/award number: N/A).

**Disclaimer** This study was commissioned and paid for by the World Health Organization (WHO). Copyright of the original work, a report submitted to WHO, belongs to WHO. This article was developed based on the WHO report and the authors have been given permission to publish. The authors alone are responsible for the views expressed in this publication and they do not necessarily represent the views, decisions or policies of WHO.

**Competing interests** None declared.

**Patient consent for publication** Not required.

**Provenance and peer review** Not commissioned; externally peer reviewed.

**Data availability statement** All data relevant to the study are included in the article or uploaded as supplementary information. No original data were generated for this study. All studies included in the evidence map are presented in the supplementary file.

**ORCID iDs**
Ani Movsisyan http://orcid.org/0000-0003-0258-8912
Jacob Burns http://orcid.org/0000-0003-4015-6862
Lisa Maria Pfadenhauer http://orcid.org/0000-0001-5038-8072
Peter von Philipsborn http://orcid.org/0000-0001-7059-6944

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
