## [Reviewer comments · BMJ Open]

ARTICLE DETAILS

TITLE (PROVISIONAL)	Travel-related control measures to contain the COVID-19 pandemic: an evidence map
AUTHORS	Burns, Jacob; Movsisyan, Ani; Biallas, Renke; Coenen, Michaela; Geffert, Karin; Horstick, Olaf; Klerings, Irma; Pfadenhauer, Lisa; von Philipsborn, Peter; Sell, Kerstin; Strahwald, Brigitte; Stratil, Jan; Voss, Stephan; Rehfuess, Eva

VERSION 1 – REVIEW

REVIEWER	Bernd Salzberger Dept. Infection Control and Inf. Diseases, Universitätsklinikum Regensburg, 93053 Regensburg
REVIEW RETURNED	10-Jul-2020

GENERAL COMMENTS	This is a well done review to map the evidence for travel restrictions on infectious disease transmission. No further comments
--

REVIEWER	Martin Falk University of South-Eastern Norway Norway
REVIEW RETURNED	30-Jul-2020

GENERAL COMMENTS	Summary: This paper summarizes and presents the non-pharmaceutical interventions related to the SARS Cov2 pandemic and related influenza in the past worldwide until early May 2020. It focuses on different types of travel-related control measures (i.e. border closures, travel restrictions and bans, entry and exit controls, quarantine/isolation of travelers when crossing borders, combined multiple interventions). The work is based on a report for the World Health Organization and includes 122 studies. These studies are summarised in Table 1 (Characteristics of included studies). The work is quite informative and gives a good overview of the intervention, and the selection criteria are carefully documented. However, the work is largely descriptive. No methods or statistical/econometric analyses are used. I also miss the bi-lateral perspective (country pairs) when considering travel restrictions. I have several suggestions. Below are a number of comments that need to be addressed. Main points The contribution of the work to literature and knowledge must be better emphasized. If the purpose of the work is to map the measures, an online database is more appropriate. Please check the Oxford Tracker below. There are already many attempts to collect non-pharmaceutical measures. https://supertracker.spi.ox.ac.uk/policy-trackers/ The authors gave an impressive literature review with detailed
---

	information. However, I would have expected that the information and data obtained would lead to a better understanding of the introduction and timing of different types of non-pharmaceutical interventions. Examples include the creation of a lockdown index such as the Woodside 2020 Table 1 trial, which classifies countries according to the strength of the lockdown regime (scaled from 1 to 10). South Korea has the least strong regime and China the strictest. Or you can analyse the introduction of various non-pharmaceutical interventions using logit/probit or count data models. The work has a static view it is a snapshot. The reality, however, is that government regulations change rapidly, often from week to week, and in some cases several times for short periods. It is therefore also necessary to gather information on the timing and duration of border controls and other interventions. The study by Summan and Nandi (2020) shows that the timing of the introduction of measures plays an important role. My other main concern is the use of secondary sources. I am not satisfied with the use of secondary sources, you should use primary sources. Entry requirements are the responsibility of government agencies (in some countries the police). This information can be obtained from the websites of the governments. Here some examples for the Nordic countries: https://politi.dk/en/coronavirus-in-denmark/travelling-in-or-out-of-denmark/entry-into-denmark https://www.fhi.no/en/op/novel-coronavirus-facts-advice/facts-and-general-advice/travel-advice-COVID19/ https://www.raja.fi/current_issues/guidelines_for_border_traffic see for example the Finnish regulations: The instructions of the Finnish Border Guard to passengers regarding entry to Finland These instructions issued by the Finnish border guard provide passengers with information on the changes that apply to entry to Finland as of 27 July. Contents  • 1. General • 2. Border traffic turned to normal  o 2. 1 Internal border traffic without restrictions: the Netherlands, Belgium, Ita-ly, Iceland, Greece, Latvia, Liechtenstein, Lithuania, Malta, Norway, Germany, Slovakia, Denmark, Hungary and Estonia o 2.2 Lifting of restrictions on external border traffic for Andorra, Cyprus, Ire-land, San Marino and the Vatican o 2.3 Lifting of restrictions on external border traffic for residents of South Ko-rea, Georgia, Japan, China, Rwanda, Thailand, Tunisia, Uruguay and New Zea-land • 3. Restriction category 1  o 3.1 Partial continuation of internal border control: Austria, Spain, Luxembourg, Portugal, Poland, France, Sweden, Slovenia, Switzerland and the Czech Re-public o 3.2 External border traffic from Bulgaria, Croatia, Romania, the United King-dom and Monaco o 3.3 Restriction category 1, permitted traffic • 4. Restriction category 2  o 4.1 Permitted traffic in restriction category 2: third countries, for example Rus-sia and the United States As you can see, there is a clear need to study bilateral entry regulations, so you need to extend the study to include country pairs. Specific comments Table 1 last column: The outcome category: number or proportions of cases is too broad and not meaningful.
--	---

	There are very specific rules now, see example for Norway “The requirements for entry quarantine do not apply for travellers who are resident in countries in the EU/EEA/Schengen area with fewer than 20 confirmed cases per 100 000 inhabitants during the last two weeks (evaluated on a national level), and fewer than 5 percent positive tests on average per week over the last two weeks.” Of course, I see that the study was completed at the beginning of May and that this information was not available at the time of the survey. I also agree that it is difficult to produce a timeless analysis. However, the criterion number of cases is not detailed enough. Future work It would be good to say whether the information obtained can be used to model the effects of the closure (as in Flaxman et al., 2020; Demirguc-Kunt, et al. 2020) or to study the determinants Demirguc-Kunt, A., Lokshin, M., & Torre, I. (2020). The sooner, the better: The early economic impact of non-pharmaceutical interventions during the COVID-19 pandemic. World Bank Policy Research Working Paper, (9257). Flaxman, S., Mishra, S., Gandy, A., Unwin, H. J. T., Mellan, T. A., Coupland, H., ... & Monod, M. (2020). Estimating the effects of non-pharmaceutical interventions on COVID-19 in Europe. Nature, 1-5. Summan, A., & Nandi, A. (2020). Timing of non-pharmaceutical interventions to mitigate COVID-19 transmission and their effects on mobility: A cross-country analysis. medRxiv. Woodside, A. G. (2020). Interventions as experiments: Connecting the dots in forecasting and overcoming pandemics, global warming, corruption, civil rights violations, misogyny, income inequality, and guns. Journal of Business Research, 117, 212-218.
--	---

REVIEWER	Dervla Kelly University of Limerick, Ireland
REVIEW RETURNED	07-Sep-2020

GENERAL COMMENTS	Overall comments: Thank you for the opportunity to review this paper. Types of outcomes of travel related interventions and types of research studies undertaken are explored in this scoping review. The review is useful for others planning to undertake research in this field. Some comments: Abstract: Line 42: “decisions”: Are the authors referring to policy decisions or research decisions? Line 42: “these challenges”: no specific challenges mentioned in previous statement. Please clarify. Suggest add a sentence to abstract to the effect that: Additional research is needed to understand their effectiveness. One of the key useful parts of the paper I think is the comprehensive list of travel related outcomes for researchers/policy makers collecting data in this area. Worth incorporating this as a strength of the paper. Introduction: Line 13: “various travel bans”: local, international or both Line 25: “Such decisions need to be based on... “: perhaps could nuisance this to reflect how policy makers are often required to take action when evidence base is weak, as recommended by WHO. Suggest: “such decisions, where possible, need to be take some decisions even if evidence base is weak...”
---

	Methods: P7 Line 5: Please add a scoping review question. Results: Figure 2: Increase text size in legend Figure 3: Couldn't interpret it. Increase text size in x axis legend. Add more info to legend and narrative or present info in a different way. Figure 4: Legends not legible Discussion: P27 Line 36: are the authors saying the COVID evidence base in this situation with respect to travel is similar to the overall evidence base with lots of pre-prints? Please update the wording to reflect this P27 Line 42: paragraph 3 of discussion is results rather than discussion. Suggest incorporate there. Did the researchers identify any gaps in which categories of interventions and which research stages have not been addressed so far in studies evaluating travel measures? This seemed to be mentioned through text. I think it would be worth summarising all the gaps in a narrative paragraph for readers.
--	---

REVIEWER	Chenyu Sun AMITA Health Saint Joseph Hospital Chicago, Chicago, IL, USA
REVIEW RETURNED	07-Oct-2020

GENERAL COMMENTS	This manuscript is overall well-written. However, two minor revisions should be noticed 1. As this work is commissioned by WHO, it would be more reasonable to search even broader databases, given it is mainly focused on COVID-19. As a matter of fact, quite a lot of original studies are not published in English or databases such as PubMed. As COVID-19 started in Wuhan, China, many original studies in China are published in Chinese journals, and China is taking more strict measures than most western countries, so it would be better if the authors could search Chinese databases CNKI, Wangfang, Vip. However, if authors have no access to those databases, or can not read them due to language barriers, I would recommend the authors to clear mention this as part of the limitations. 2. Although this work is mainly focused on Travel-related control measures, confounding factors can not be neglected. I suggest discussing a little more of theses in the manuscript. For example, in one systemic review and meta-analysis on the protective effect of wearing masks, mask-wearing during the flight was also discussed briefly. Liang M, Gao L, Cheng C, Zhou Q, Uy JP, Heiner K, Sun C. Efficacy of face mask in preventing respiratory virus transmission: A systematic review and meta-analysis. Travel Med Infect Dis. 2020 Jul-Aug;36:101751. doi: 10.1016/j.tmaid.2020.101751. In summary, I recommend this manuscript to be accepted after a minor revision of the two points mentioned above.
---

VERSION 1 – AUTHOR RESPONSE

Reviewer 1			
	This is a well done review to map the evidence for travel restrictions on infectious disease transmission. No further comments.	No changes.	We thank the reviewer for their positive feedback.
Reviewer 2			
Aims/introduction	The contribution of the work to literature and knowledge must be better emphasized. If the purpose of the work is to map the measures, an online database is more appropriate. Please check the Oxford Tracker below. There are already many attempts to collect non-pharmaceutical measures. https://supertracker.spi.ox.ac.uk/policy-trackers	Please, see revisions in the Abstract and in the last paragraph of the Introduction.	We thank the reviewer for their comments. As also requested by the Associate Editor and Reviewer 3, we have now clearly specified the aim of this scoping review/evidence map and have highlighted explicitly in the abstract and the text that this review aims to outline the elements of an evidence base assessing the impact of travel-related control measures to contain the COVID-19 pandemic in terms of the study designs, specific measures and outcomes assessed. Such an evidence map can then inform future research and evidence syntheses on the effectiveness of these measures. While we agree with the reviewer that living tools such as the cited online platform are very informative and

			provide a dynamic output, that was beyond the scope of our work, and is indeed beyond the scope of the established evidence map methodology.
Discussion	The authors gave an impressive literature review with detailed information. However, I would have expected that the information and data obtained would lead to a better understanding of the introduction and timing of different types of non-pharmaceutical interventions. Examples include the creation of a lockdown index such as the Woodside 2020 Table 1 trial, which classifies countries according to the strength of the lockdown regime (scaled from 1 to 10). South Korea has the least strong regime and China the strictest. Or you can analyse the introduction of various non-pharmaceutical interventions using logit/probit or count data models.	Please, see Figure 4, online supplementary file S6 and revisions in the Implications for Moving Forward section of the Discussion.	We agree with the reviewer that additional data on the introduction and timing of different travel-related control measures (note: broader ‘NPIs’, as described by the reviewer, were beyond the scope of this evidence map), would have been helpful in better understanding the existing measures. Most studies did not report on these aspects consistently or in detail; based on what was available in the reported studies, we coded and presented in Figure 4 the phases of an epidemic or pandemic during which the measure was implemented (e.g., early, local transmission and post-peak). Furthermore, we presented more details on the specific interventions and their phase of

			implementation in the online supplementary file S6. We have now revised the text to better signpost this important output. Finally, we highlighted the need to better capture these aspects through consideration of data from external non-scientific sources in the Implications for Moving forward section of the Discussion. Regarding the last suggestion on logit/probit or count data models, such an examination of association between severity of lockdown and the resulting transmission outcomes is outside the scope of this evidence map, and in general the methodology of evidence maps.
Discussion	The work has a static view it is a snapshot. The reality, however, is that government regulations change rapidly, often from week to week, and in some cases several times for short periods. It is therefore also necessary to gather information on the timing and duration of border controls and other interventions. The study by Summan and Nandi (2020) shows that the timing of the introduction of measures plays an important role.	Please, see revisions in the Implications for Moving Forward section of the Discussion.	We appreciate the reviewer's comment on the need to consider additional information in relation to the implementation (e.g., timing and duration) of the measures. As described above, where data on timing was reported

			in included studies, we did consider this. We have also now highlighted this point for future research and synthesis in the Implications for Moving Forward section of the Discussion.
Methods	My other main concern is the use of secondary sources. I am not satisfied with the use of secondary sources, you should use primary sources. Entry requirements are the responsibility of government agencies (in some countries the police). This information can be obtained from the websites of the governments.... As you can see, there is a clear need to study bilateral entry regulations, so you need to extend the study to include country pairs.	Please, see revisions in the Implications for Moving Forward section of the Discussion.	The aim of this evidence map was to identify and document studies that have assessed the impact of travel-related control measures. We did not consider external sources in our map, such as governmental websites, because of the short timeframe, and because it did not immediately pertain to the scope of the evidence map. We think this is a good point, however, and highlighted this for future research and synthesis in the Implications for Moving Forward section of the Discussion.
Results/outcomes	Table 1 last column: The outcome category: number or proportions of cases is too broad and not meaningful. There are very specific rules now, see example for Norway “The requirements for entry quarantine do not apply for travellers who are resident in countries in the EU/EEA/Schengen area with fewer than 20 confirmed cases per 100 000 inhabitants during the last two weeks (evaluated on a national level), and fewer than 5 percent positive tests on	Please, see revisions in the Outcome Categories and Outcomes section in the Results.	We agree with the reviewer that the mentioned category is very broad. However, this category was derived based on the specific outcomes reported in the included studies. In the online

	average per week over the last two weeks. “” Of course, I see that the study was completed at the beginning of May and that this information was not available at the time of the survey. I also agree that it is difficult to produce a timeless analysis. However, the criterion number of cases is not detailed enough.		supplementary file S7 we provide further details regarding the specific outcomes reported under this broad category, including number of cases as rates (e.g., number per 100.000) as well as how often these were used in the included studies. We have revised the text in the Results to better describe and signpost this.
Future work	It would be good to say whether the information obtained can be used to model the effects of the closure (as in Flaxman et al., 2020; Demirguc-Kunt, et al. 2020) or to study the determinants	No changes.	We appreciate the comment, and agree that the Flaxman study was very informative regarding the impact of NPIs; however, we would suggest not adding this additional information. In such a modelling study, how NPIs are modelled is a critical decision, and will depend upon a range of factors, most importantly the specific research question of the study. We feel that such information would be beyond the scope of this evidence map, which involved assessing the research from a broad perspective, not looking towards individual primary modelling studies.

Reviewer 3			
Abstract	Line 42: "decisions": Are the authors referring to policy decisions or research decisions? Line 42: "these challenges": no specific challenges mentioned in previous statement. Please clarify.	Please, see the revised Conclusions section in the Abstract.	We originally referred to 'research decisions' and 'challenges in relation to the diversity of methods used', but have now revised the entire sentence to make it clear and concise.
Abstract	Suggest add a sentence to abstract to the effect that: Additional research is needed to understand their effectiveness.	Please, see the revised Conclusions section in the Abstract.	As the Associate Editor also had a suggestion on this, we have revised the Conclusions to highlight that this map is not sufficient to assess the effectiveness of different measures and that it outlines specifics of the evidence base that could inform future research and evidence synthesis.
Abstract	One of the key useful parts of the paper I think is the comprehensive list of travel related outcomes for researchers/policy makers collecting data in this area. Worth incorporating this as a strength of the paper.	No changes.	We very appreciate the reviewer's feedback. While we agree with the reviewer, the word limit for the abstract (max: 300, currently: 300) unfortunately does not allow us to describe the strengths and limitations of the paper.
Introduction	Line 13: "various travel bans": local, international or both	Please, see revisions in the second paragraph of the introduction.	We meant both and have now specified it in the introduction.

Introduction	Line 25: "Such decisions need to be based on... ": perhaps could nuisance this to reflect how policy makers are often required to take action when evidence base is weak, as recommended by WHO. Suggest: "such decisions, where possible, need to be take some decisions even if evidence base is weak... "	Please, see revisions in the second paragraph of the introduction.	We thank the review for this suggestion and have made revisions to incorporate it.
Methods	P7 Line 5: Please add a scoping review question.	Please, see revisions in the last paragraph of the Introduction.	As also requested by the Associate Editor, we have added a specific aim for our paper in line with the broad goal of a scoping review to the end of the Introduction.
Results	Figure 2: Increase text size in legend	Please see adapted Figure 2	We have increased the size of the text in the legend
Results	Figure 3: Couldn't interpret it. Increase text size in x axis legend. Add more info to legend and narrative or present info in a different way.	Please see adapted Figure 3	We have adapted the text in the legend and axis labels so that it is more legible
Results	Figure 4: Legends not legible	Please see adapted Figure 4	We have adapted the legend and axis labels in the figure so that it is more legible.
Discussion	P27 Line 36: are the authors saying the COVID evidence base in this situation with respect to travel is similar to the overall evidence base with lots of pre-prints? Please update the wording to reflect this	Please, see revisions in the second paragraph of the Summary of Findings section in the Discussion.	We apologise for this confusion. We have revised the wording to add clarity.
Discussion	P27 Line 42: paragraph 3 of discussion is results rather than discussion. Suggest incorporate there.	No changes.	We appreciate the reviewer's suggestion. However, we would request to leave this section in the Summary of Findings section of the Discussion, as it provides a summary

			of key regions covered by the included studies. The descriptive data for these regions are presented in the Countries and Settings section of the Results.
Discussion	Did the researchers identify any gaps in which categories of interventions and which research stages have not been addressed so far in studies evaluating travel measures? This seemed to be mentioned through text. I think it would be worth summarising all the gaps in a narrative paragraph for readers.	Please, see Gaps in the Current Evidence Base section in the Discussion.	As suggested by the reviewer, we have now summarised all the key gaps in a separate paragraph.
Reviewer 4			
Discussion	As this work is commissioned by WHO, it would be more reasonable to search even broader databases, given it is mainly focused on COVID-19. As a matter of fact, quite a lot of original studies are not published in English or databases such as PubMed. As COVID-19 started in Wuhan, China, many original studies in China are published in Chinese journals, and China is taking more strict measures than most western countries, so it would be better if the authors could search Chinese databases CNKI, Wangfang, Vip. However, if authors have no access to those databases, or cannot read them due to language barriers, I would recommend the authors to clear mention this as part of the limitations.	Please, see revisions in the Methodological Limitations of this Study section in the Discussion.	We thank the reviewer for this comment. We have now explicitly highlighted this as part of the limitations.
Discussion	Although this work is mainly focused on Travel-related control measures, confounding factors cannot be neglected. I suggest discussing a little more of these in the manuscript. For example, in one systemic review and meta-analysis on the protective effect of wearing masks, mask-wearing during the flight was also discussed briefly.	Please, see revisions in the second paragraph of the Implications for Moving Forward section of the Discussion.	We agree with the reviewer, that to contain the pandemic, governments have employed a range of public health measures, which challenges assessment of the effectiveness of a single measure raising the issue of potential confounding. We

			have now expanded the discussion on this issue in the Implications for Moving Forward section and have highlighted a few suggestions for future research to address this.
--	--	--	---

VERSION 2 – REVIEW

REVIEWER	Martin Falk University of South-Eastern Norway (USN)
REVIEW RETURNED	25-Nov-2020

GENERAL COMMENTS	Summary: The authors have addressed many of the points, and the paper reads well. I also appreciate the efforts to collect all these studies and categorize all the information. However, the main approach is the same, and the sources for travel bans and restrictions are based on secondary sources rather than on primary and official sources. Although there is a nice attempt to visualize the measures, the main weakness of the study remains that it is a simple mapping exercise without any attempt to analyze the data. Main comments Summary of the results: The mapping of the measures does not lead to conclusions. This is all vague and superficial. For example, we learn from the study that travel-related control measures probably vary between the different phases of the pandemic. Such conclusions cannot be drawn because one has to compare the number of Covid-19 cases or deaths or patients in the intensive care unit with the measures. Another conclusion of the study is that reporting on the timing of implementation tends to be poor. This is a weakness of the secondary sources. Information is available on government websites, and you can use the Wayback website approach to get information from earlier days. I still believe that the approach collecting information from secondary sources is wrong. The government (following the suggestions from their expert agencies such as the national health authorities) is responsible for border closures, entry screening and travel restrictions. You have to use the primary official sources (government, national health institutions). I am not convinced that the mapping approach is useful. The approach for the Covid-19 pandemic is static, while the government measures are very volatile and dynamic. The duration of the measures would be interesting. More work is needed here. The methods here are limited to descriptive statistics. More can be done with the data such as box plots, statistical tests and correlations. There is no attempt to analyse the data. Specific comments P1 “Following the early-stage responses in Asian countries, strict measures, such as border closures and drastic reductions in airline travel, have been put into place in most countries around the world,
--

	starting in February 2020 and continuing into May 2020.” Reference is needed here. There are other examples in the paper.
REVIEWER	Dervla Kelly University of Limerick, Ireland
REVIEW RETURNED	18-Nov-2020
GENERAL COMMENTS	Dear Authors, Thank you for the revisions on the manuscript. My comments have been addressed. Well done on the manuscript.
REVIEWER	Chenyu Sun AMITA Health Saint Joseph Hospital Chicago
REVIEW RETURNED	12-Nov-2020
GENERAL COMMENTS	No need for further revisions. Ready for publication